# Clonal Diversity of *Klebsiella* spp. and *Escherichia* spp. Strains Isolated from Patients with Ventilator-Associated Pneumonia

**DOI:** 10.3390/antibiotics10060674

**Published:** 2021-06-05

**Authors:** Jan Papajk, Kristýna Mezerová, Radovan Uvízl, Taťána Štosová, Milan Kolář

**Affiliations:** 1Department of Anesthesiology, Resuscitation and Intensive Care Medicine, University Hospital Olomouc, 77900 Olomouc, Czech Republic; jan.papajk@fnol.cz (J.P.); radovan.uvizl@fnol.cz (R.U.); 2Department of Microbiology, Faculty of Medicine and Dentistry, Palacký University Olomouc, 77515 Olomouc, Czech Republic; 3Department of Microbiology, University Hospital Olomouc, 77515 Olomouc, Czech Republic; tatana.stosova@fnol.cz (T.Š.); milan.kolar@fnol.cz (M.K.)

**Keywords:** ventilator-associated pneumonia, *Klebsiella* spp., *Escherichia* spp., pulsed-field gel electrophoresis (PFGE), clonality, endogenous infection

## Abstract

Ventilator-associated pneumonia (VAP) is one of the most severe complications affecting mechanically ventilated patients. The condition is caused by microaspiration of potentially pathogenic bacteria from the upper respiratory tract into the lower respiratory tract or by bacterial pathogens from exogenous sources such as healthcare personnel, devices, aids, fluids and air. The aim of our prospective, observational study was to confirm the hypothesis that in the etiology of VAP, an important role is played by etiological agents from the upper airway bacterial microflora. At the same time, we studied the hypothesis that the vertical spread of bacterial pathogens is more frequent than their horizontal spread among patients. A total of 697 patients required mechanical ventilation for more than 48 h. The criteria for VAP were met by 47 patients. Clonality of bacterial isolates from 20 patients was determined by comparing their macrorestriction profiles obtained by pulsed-field gel electrophoresis (PFGE). Among these 20 patients, a total of 29 PFGE pulsotypes of *Klebsiella* spp. and *Escherichia* spp. strains were observed. The high variability of clones proves that there was no circulation of bacterial pathogens among hospitalized patients. Our finding confirms the development of VAP as a result of bacterial microaspiration and therefore the endogenous origin of VAP.

## 1. Introduction

Ventilator-associated pneumonia (VAP) is one of the most severe complications affecting mechanically ventilated patients. VAP has a considerable impact on patient prognosis due to longer ventilation time and higher mortality. The incidence of VAP is reported to be 9–27% [1]. The risk of developing VAP is highest in the first five days of mechanical ventilation and then gradually decreases [2]. Mortality associated with VAP is 33–50%, depending on the bacterial pathogen, adequacy of the antibiotic therapy and comorbidities of the patient [1]. With strict preventive measures in place, the death rate may decrease to approximately 9–13% [3]. The incidence of VAP is influenced by a range of risk factors contributing to undesirable microaspiration of potentially pathogenic bacteria into the lower respiratory tract. Microaspirations occur not only due to respiratory invasive procedures, including intubation or bronchoscopy, but also passively due to leakage of pooled secretions (from the paranasal sinuses, nasopharynx, oropharynx or stomach) from the space above the endotracheal tube cuff. Even insertion of the tube itself disrupts the natural barrier protecting the airways [4] and a biofilm forming on its surface may become a source of bacterial contamination of the lower respiratory tract [5]. All these mechanisms, individually or together, participate in the endogenous origin of VAP. The independent risk factors for VAP are mainly previous administration of antibiotics and patient comorbidities [1]. The exogenous sources of bacterial pathogens responsible for VAP in the intensive care setting may include healthcare personnel, devices, aids, fluids or air [6].

The study aimed to confirm the hypothesis that in the etiology of VAP, an important role is played by etiological agents from the upper airway bacterial flora, and that vertical spread of bacterial pathogens is more common than their horizontal spread among patients.

## 2. Results

Patients over 18 years of age hospitalized at the Department of Anesthesiology, Resuscitation and Intensive Care Medicine, University Hospital Olomouc, Czech Republic, between 11 January 2018 and 22 May 2020 were involved in this prospective, observational study.

Over the study period, a total of 697 patients required mechanical ventilation for more than 48 h. The criteria for VAP were met by 47 patients. The clonality study included 20 patients (18 males, 2 females) whose VAP was caused by *Klebsiella* spp. and *Escherichia* spp. As for demographic data, only age was normally distributed (median 58, interquartile range, IQR, 23–83). The other parameters showed non-normal distribution. The mean body mass index was 28 ± 5 (median 28, IQR 19–48). The median Acute Physiology and Chronic Health Evaluation II (APACHE II) score on admission was 24 (IQR 14–32). The median duration of mechanical ventilation and intensive care unit (ICU) stay was 11 days (IQR 5–43) and 13 days (IQR 6–44), respectively. The patients had a 30-day mortality rate of 20%.

A total of 49 *Klebsiella* spp. and *Escherichia* spp. isolates were obtained from 20 patients. Of those, nine *Escherichia* spp. isolates came from five patients and 40 *Klebsiella* spp. isolates from 17 patients; the latter included four isolates from patients from whom *Klebsiella variicola* or *Escherichia* spp. isolates were also obtained.

### 2.1. PFGE Analysis of Klebsiella spp. Isolates

For 26 *Klebsiella pneumoniae* isolates from 12 patients, different restriction profiles obtained with SpeI digestion of DNAs were observed, resulting in 16 pulsotypes. Isolates with identical restriction profiles were documented among different types of samples collected from the patients (Figure 1). Additionally, more than one clone was observed among isolates coming from patients referred to as PAT 8, PAT 9 and PAT 12.

Clustering analysis of nine isolates identified as *Klebsiella variicola* coming from five patients contributed to our finding that there was no clonal spread among patients in the Department of Anesthesiology, Resuscitation and Intensive Care Medicine. Furthermore, more than one clone was observed in the patient PAT 2 (Figure 2). A dendrogram constructed for five *Klebsiella oxytoca* isolates coming from 2 patients supported clonal diversity among different patients (Figure 3). Nevertheless, the phylogenetic tree based on PFGE profiles revealed identical strains isolated from different types of samples collected from each patient.

### 2.2. PFGE Analysis of Escherichia spp. Isolates

PFGE identified three pulsotypes among five *Escherichia coli* isolates belonging to three patients (Figure 4), supporting clonal diversity among different patients. The same results were obtained by restriction profiles of *Escherichia hermannii* DNA as four isolates were categorized to two phylotypes corresponding with patient codes (Figure 5).

In 15 patients (75%), the bacterial pathogen that caused VAP was isolated from the upper respiratory tract before the development of VAP, or bacterial strains identified from the upper respiratory tract and endotracheal aspirate (ETA) samples showed identical restriction profiles in the same patient. In the remaining five patients, we confirmed the presence of the detected pathogen only in ETA samples after VAP had occurred.

## 3. Discussion

Over the study period, the incidence of VAP in the Department of Anesthesiology, Resuscitation and Intensive Care Medicine reached 7%, a lower rate compared to long-term reports [1]. The incidence was compared to 2018 data (1 January–31 December 2018) showing 20 cases of VAP among 291 patients mechanically ventilated for over 48 h (unpublished data). The 2018 rate was also 7%, suggesting a stable incidence of VAP in the department. In their comparison of data from the National Healthcare Safety Network, Dudeck et al. reported a downward trend in the incidence of VAP [7,8]. However, recent guidelines by the Infectious Diseases Society of America and the American Thoracic Society state a stable VAP rate (approx. 10%) in mechanically ventilated patients [9]. The present study found a mortality rate of 20% among VAP patients. A meta-analysis by Melsen et al. reported lower overall mortality (13%). However, the opposite was true for comparable patient subgroups (APACHE II score 20–29), with rates of 20% and 36%, respectively [3]. Mortality rates in different studies are also difficult to compare due to various methodologies. According to Forel et al., 28-day and 90-day mortality of patients with VAP was 27% and 42%, respectively [10]. While a study from Thailand documented 30-day mortality of 46% [11], Tejada et al. reported a rate of 44% without time specification [12]. A multi-center study by Herkel et al. assessed 30-day mortality with regard to adequate or inadequate empirical antibiotic therapy (27% vs. 45%) [13].

In the pathogenesis of VAP, microaspiration of bacteria from secretions in the space above the endotracheal tube cuff or the oropharynx plays the major role. Less frequently, the infection is caused by exogenously acquired pathogens. Thus, most VAP cases are of endogenous origin [6,14]. The risk factors for the development of VAP are mainly urgent tracheal intubation, reintubation, bronchoscopy and intolerance of enteral feeding, that is, situations representing a higher risk of microaspiration [15]. The introduction of routine assessment of the bacterial flora in both the upper and lower respiratory tract has allowed early identification of bacterial pathogens responsible for VAP. ETA sample collection results in more frequently detected pathogens (93%) compared to not only oropharyngeal swab (OS) and gastric aspiration samples (59% and 57%, respectively), but especially to protected specimen brushing (36%) [16]. The latter approach has the lowest sensitivity of all invasive techniques and is therefore not routinely performed, as stated in a meta-analysis by Fernando et al. [17]. Our results suggest that taking OS and ETA samples twice a week is sufficient.

To the best of our knowledge, this is the first study comparing the clonality of *Klebsiella* and *Escherichia* spp. strains isolated from different types of samples in each patient with VAP. Among 20 patients, a total of 29 PFGE pulsotypes were observed. Restriction profiles obtained from 26 *K. pneumoniae* isolates revealed 16 pulsotypes among 12 patients. *K. variicola* isolates also showed high variability of clones, as six pulsotypes were obtained from isolates belonging to five patients. Where *K. oxytoca*, *E. coli* and *E. hermannii* isolates were analyzed by PFGE, the number of pulsotypes always corresponded with the number of patients, proving that there was no circulation of pathogen among hospitalized patients.

Comparison of PFGE pulsotypes obtained in this study also showed high variability of bacterial clones among different patients, which correlates with a study by Hanulik et al. demonstrating clonal diversity among 23 *K. pneumoniae* isolates with only three pairs of clones belonging to six different patients [18]. A later study by Pudova et al. observed unique restriction profiles of analyzed isolates (74%) from hospital-acquired pneumonia (HAP) patients, suggesting only rare clonal spread of HAP bacterial pathogens among individual patients [19]. Unlike our study, circulation of multidrug-resistant *K. pneumoniae* isolates causing VAP in Egypt was confirmed by PFGE analysis. A study by Mohamed et al. included 19 clinical isolates from different patients and four environmental isolates, resulting in 17 pulsotypes. The restriction profiles of clinical and environmental isolates were similar, suggesting the exogenous origin of VAP infection [20]. Identical clones of *K. pneumoniae* were also detected in 29% of VAP patients in China. Clonal relationship of the *K. pneumoniae* strains was studied by random amplified polymorphic DNA (RAPD) and multilocus sequence typing, revealing 21 different RAPD patterns and 25 sequence types (ST) among 49 *K. pneumoniae* isolates. The most frequent clone, ST23 (43% prevalence) corresponding to hypervirulent *K. pneumoniae* (hvKP), evinced an identical RAPD pattern for all analyzed isolates, suggesting their epidemiological relationship [21]. According to a study by Tabrizi et al., hvKP isolated from mechanically-ventilated drug-poisoning patients was related to ST23 with the same PFGE pulsotype pattern, also underlining its epidemiological importance [22]. Based on our PFGE results, no epidemiologic strain was detected among the investigated patients.

In 75% of patients, the identified bacterial pathogen was first isolated from the upper respiratory tract as part of the bacterial flora. Subsequently, its presence in the endotracheal secretion of a patient with clinically expressed VAP was confirmed. This finding confirms the development of VAP resulting from bacterial microaspiration as reported by Pudova et al. [19].

The study limitations are its retrospective design, evaluation of only two bacterial species, and inclusion of only one center.

## 4. Materials and Methods

### 4.1. Patients and Clinical Material

All patients over 18 years of age who required mechanical ventilation for more than 48 h at the Department of Anesthesiology, Resuscitation and Intensive Care Medicine over the study period were involved in the study. Biological samples for microbiology testing, collected as described below, were a routine part of standard care for mechanically ventilated patients.

### 4.2. Sample Processing

The study included patients who met the clinical criteria for VAP caused by strains of *Klebsiella* spp. and *Escherichia* spp. The clinical signs of pneumonia are defined as the presence of newly developed or progressive infiltrates on chest radiographs plus at least two other signs of respiratory tract infection: temperature > 38 °C, purulent sputum, leukocytosis (white blood cell, WBC > 10 × 10^3^/mm^3^) or leukopenia (WBC < 4 × 10^3^/mm^3^), signs of inflammation on auscultation, cough and/or respiratory insufficiency with a PaO_2_/FiO_2_ ratio of ≤300 mm Hg that develop after more than 48 h of mechanical ventilation.

Excluded were patients with signs of pneumonia at the time of initiation of mechanical ventilation or with neutropenia, organ donors, those who later switched to withhold therapy or palliative care, and patients with VAP caused by bacterial pathogens other than *Klebsiella* spp. and *Escherichia* spp.

### 4.3. Collection of Samples for Microbiological Culture and Identification of Isolated Bacteria

In all patients, OS samples were taken on admission and then twice weekly, and ETA samples were taken twice weekly following the initiation of mechanical ventilation. OS samples were collected from the back wall of the oropharynx using a commercially available sample collection kit with a transport medium (Copan Diagnostics, Murrieta, CA, USA). ETA samples were collected by aspiration of secretions from an orotracheal tube using a sterile closed collecting system, with subsequent rinsing of the suction catheter with 10 mL of sterile saline and closing of the test tube with a sterile stopper. The identification of bacteria was performed by MALDI-TOF MS (Biotyper Microflex, Bruker Daltonics, Bremen, Germany) [23]. The study analyzed all *Escherichia* spp. and *Klebsiella* spp. strains isolated from OS and ETA collected from the enrolled patients. Appendix A summarizes isolated bacterical species, dates of sample collection and VAP diagnosis of the enrolled patients.

### 4.4. Genotyping of Selected Bacterial Isolates

Clonal relationships of bacterial isolates were determined by comparing their macrorestriction profiles obtained by pulsed-field gel electrophoresis (PFGE). Bacterial DNA was isolated according to a protocol previously described in a study by Husickova et al. [24]. Briefly, DNA was isolated from a bacterial culture incubated at 37 °C for 16 h in Mueller–Hinton broth. The cells were washed three times with washing buffer followed by dilution of bacteria for optical density of 1.0–1.5 at 600 nm. The cells were subsequently mixed with 2% low melting point agarose (Bio-Rad, Hercules, CA, USA) to form blocks. The prepared agarose blocks were incubated at 37 °C in lysis buffer containing lysozyme. After 24 h, the blocks were transferred into deproteination buffer with proteinase K and incubated at 55 °C for another 24 h. The obtained blocks were finally washed in TE buffer and stored at 4 °C. According to manufacturer’s instructions, enzymes XbaI and SpeI (New England Biolabs, Ipswich, MA, USA) were used for DNA restriction. *Escherichia* spp. isolates were digested by XbaI while SpeI-digested genomic DNA was used for obtaining restriction profiles of *Klebsiella* spp. isolates. The obtained DNA fragments were separated by PFGE, which was carried out in 1.2% agarose gel at 6 V cm^−1^ and pulse times of 2–35 s for 24 h. Thiourea (final concentration 50 μM) was added to an electrophoresis buffer in order to improve the typeability of the strains. Ethidium bromide was used for staining the gel, the obtained restriction profiles were documented by an imaging device and compared by the GelCompar II software (Applied Maths, Kortrijk, Belgium). Dendrograms were constructed by the unweighted pair group method with arithmetic mean combined with a hierarchical clustering method with Dice’s coefficient. Optimization and band matching tolerance was set at 2%. Restriction profiles reaching 97% similarity were considered identical.

### 4.5. Statistical Analysis

Data were analyzed with IBM SPSS Statistics for Windows version 23.0 (IBM Corp., Armonk, NY, USA). Quantitative variables are presented as means and standard deviations (SD), minimum and maximum values and medians. The Shapiro–Wilk test for normality was used to verify that most data are not normally distributed (the only exception being age). Qualitative data (males/females, 30-day mortality) are presented as counts and percentages.

## 5. Conclusions

The results of this study, namely unique profiles confirming high variability of bacterial pathogens isolated from different patients, led us to conclude that the isolates did not spread among the hospitalized patients, suggesting that all VAP infections were endogenously acquired. At the same time, the results prove that strict adherence to hygiene and epidemiological precautions and careful surveillance of bacterial pathogens causing VAP, including assessment of the upper respiratory tract microflora, in clinical practice, may effectively prevent the horizontal spread of bacterial pathogens among mechanically ventilated patients. The observations of the study also indicate that bacterial microflora of the upper respiratory tract might predict the pathogens implicated in VAP. Thus, regular screening of a patient’s oral microflora might guide empirical antibiotic treatment when the patient subsequently develops VAP, possibly increasing the rate of appropriate antibiotic treatment.

## Figures and Tables

**Figure 1 antibiotics-10-00674-f001:**
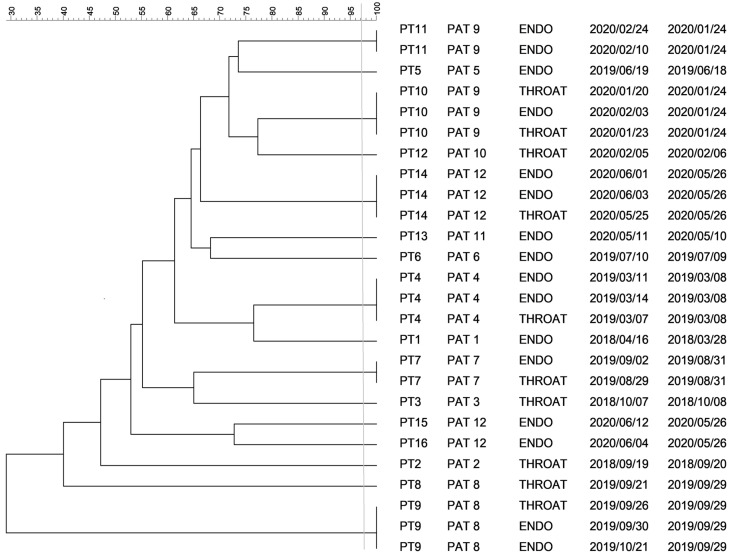
Dendrogram for restriction patterns of *K. pneumoniae* isolates. Legend: horizontal axis—similarity of isolates (%); vertical axis—pulsotype, patient code, type of sample, date of sample collection, date of VAP diagnosis; PT pulsotype, PAT patient, ENDO endotracheal aspirate sample, THROAT throat swab. The vertical line indicates the similarity threshold set at 97%.

**Figure 2 antibiotics-10-00674-f002:**
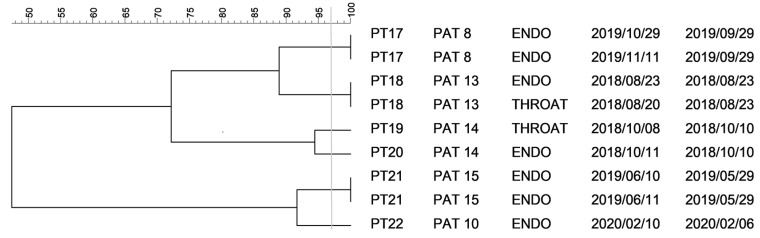
Dendrogram for restriction patterns of *K. variicola* isolates. Legend: horizontal axis—similarity of isolates (%); vertical axis—pulsotype, patient code, type of sample, date of sample collection, date of VAP diagnosis; PT pulsotype, PAT patient, ENDO endotracheal aspirate sample, THROAT throat swab. The vertical line indicates the similarity threshold set at 97%.

**Figure 3 antibiotics-10-00674-f003:**
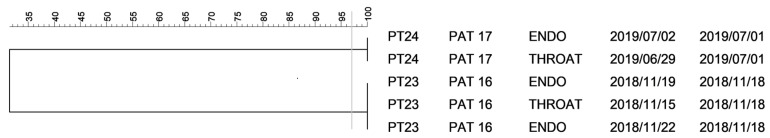
Dendrogram for restriction patterns of *K. oxytoca* isolates. Legend: horizontal axis—similarity of isolates (%); vertical axis—pulsotype, patient code, type of sample, date of sample collection, date of VAP diagnosis; PT pulsotype, PAT patient, ENDO endotracheal aspirate sample, THROAT throat swab. The vertical line indicates the similarity threshold set at 97%.

**Figure 4 antibiotics-10-00674-f004:**
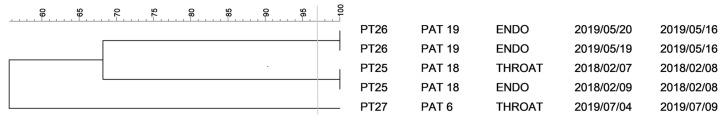
Dendrogram for restriction patterns of *E. coli* isolates. Legend: horizontal axis—similarity of isolates (%); vertical axis—pulsotype, patient code, type of sample, date of sample collection, date of VAP diagnosis; PT pulsotype, PAT patient, ENDO endotracheal aspirate sample, THROAT throat swab. The vertical line indicates the similarity threshold set at 97%.

**Figure 5 antibiotics-10-00674-f005:**
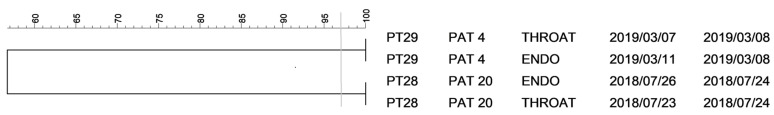
Dendrogram for restriction patterns of *E. hermannii* isolates. Legend: horizontal axis—similarity of isolates (%); vertical axis—pulsotype, patient code, type of sample, date of sample collection, date of VAP diagnosis; PT pulsotype, PAT patient, ENDO endotracheal aspirate sample, THROAT throat swab. The vertical line indicates the similarity threshold set at 97%.

## Data Availability

The data presented in this study are available on request from the corresponding author. The data are not publicly available due to protection of privacy of patients included in the study.

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
