# Peer review of "Clonal Diversity of Klebsiella spp. and Escherichia spp. Strains Isolated from Patients with Ventilator-Associated Pneumonia"

_antibiotics, 2021, doi:10.3390/antibiotics10060674_

Round 1

Reviewer 1 Report

Dear Editor and Authors, 

I have read with interest the paper by Papajk et al about "Clonal diversity of pathogenic enterobacteria isolated from patients with ventilator-associated pneumonia". In their monocentre study the authors describe the clonal diversity of Klebsiella spp. and Escherichia spp. isolates from taptients undergoing mechanical ventilation diagnosed with VAP, suggesting the endogenous origin of VAP rather than an exogenous one.

I feel that the topic addressed is of interest to the reader, and the conclusions drawn are consistent with data reported.

I would suggest some minor revisions:

  • I would move the "Methods" section right after the Introduction. In section 4.1,  suggest clarifying whether the study was approved by the Local Ethics Committee (despite no informed consent could be signed by sedated patients). In section 4.2 I suggest adding the temporal criterion (>=48h after intubation) in the definition of VAP.
  • In the "Results" section I would recommend substitutiong "abnormal distribution" (line 64) with "non-normal distribution". Moreover, in lines 64-65, I don't understand to what "respectively" refers. I recommend revising these lines so they can be more understandable.
  • In the "discussion" session, I would change "insignificantly" (line 155) with "not statistically significant". Moreover, I feel pivotal the need for a paragraph regarding the limitations of the present study (retrospective, non generalizable, only two pathogens addressed, wide time span from culture of the different isolates, etc). On the other hand, I would also suggest adding some strength points of the study: from a clinical point of view, the observations of the study indicate that upper respiratory flora might predict the pathogens implicated in low respiratory infections. Thus, regular screening of patients oral flora might guide empirical antibiotic treatment when a patient subsequently develops VAP, possibly increasing the rate of early appropriate treatments.

Given all the overall considerations, I would recommend acceptance of the paper, after minor revisions.

Author Response

Dear reviewer

Thank you very much for your efforts and for your comments and suggestions.

We have found them very useful to amend the quality and comprehensibility of our manuscript and have accepted most of them. Below we summarise all the revisions done point by point.

To make the comments and responses easier to read, the reviewer’s comments are written in italics, followed by our responses in normal font in indented blocks.

With best wishes

Kristýna Mezerová and Jan Papajk

Reviewer 2 Report

Should be published in Antibiotics after minor revision

In their manuscript entitled “Clonal diversity of pathogenic enterobacteria isolated from patients with ventilator-associated pneumonia”, Jan Papajk et al. report the result of a prospective observational study aiming to characterize the etiology of ventilator-associated pneumonia, notably to distinguish between vertical and horizontal origin of Klebsiella spp. and Escherichia spp. strains. This study highlights that good healthcare practices allow to prevent the horizontal spread of bacterial pathogens and further supports the dominant endogenous origin of VAP.

General appreciation

In my opinion, this paper is well-written and pleasant to read. Although the originality of this study is not high, it provides an up-to-date situation of the etiology of VAP at the University hospital Olomouc, Czech Republic, with observations similar to those reported in previous studies conducted in other clinical settings. I have only a few comments that should be considered for addressing minor issues. Accordingly, I recommend a minor revision of this manuscript before publication in Antibiotics.

Minor comments

The interest of this study is to provide an update of the prevailing situation for a given location and period. Accordingly, the title of the article should be completed indicating the corresponding information.

The study period is indicated in Materials and Methods line 209. It should be indicated earlier in Results, line 59.

A Table summarizing all relevant information and results for each of the 20 patients could be informative and enhance readability.

Abbreviations should be defined ‘at the right place’ in the manuscript. For instance, “OS” and “ETA” are defined lines 231 and 233 although they have been used earlier in the manuscript, from lines 163 and 113, respectively. “ICU” is not defined line 65.

As for Figures, the indication of the bacterial species is useless since each Figure corresponds to a single species that is specified in the caption; thus the corresponding column should be deleted. On the other hand, it could be useful to give a number to the different pulsotypes identified (e.g. PT#). Beside this, instead given as a “legend”, I would prefer having the corresponding information above each item in the Figure. The formatting of dates should also be checked with respect to the Journal guidelines. A vertical line may be added in dendrograms to indicate the similarity threshold (set at 97%). The suspected upper respiratory tract origin of VAP in many patients may be better evidenced/highlighted (e.g. with an arrow from “THROAT” to “ENDO”). I wonder whether patient ID refer to same patients across Figures (For example, does “PAT 1” correspond to the same patient in Figures 1-5?!; I don't think so; to avoid confusion, each patient must be given a unique number).

In Discussion, results are really discussed only from the 4th paragraph. Previous paragraphs provide interesting information but they are quite lengthy.

Some typo errors must be corrected. For instance, line 223: “O2” should be written “O2” (2 must be written as a subscript). The ratio has no unit (delete “mm Hg”).

Line 212: the lack of informed consent sounds unexpected to me. I would expect relatives being informed and asked to provide this if the patient themselves were not able to.

Line 261: please justify the threshold of 97% similarity to consider restriction profiles being identical.

Line 256: check the word “typeability”

Line 268: check the sentence (qualitative or quantitative?)

Author Response

Dear reviewer

Thank you very much for your efforts and for your comments and suggestions.

We have found them very useful to amend the quality and comprehensibility of our manuscript and have accepted almost all of them. Below we summarise all the revisions done point by point.

To make the comments and responses easier to read, the reviewer’s comments are written in italics, followed by our responses in normal font in indented blocks.

With best wishes

Kristýna Mezerová and Jan Papajk
